# Anthocyanins from *Opuntia ficus-indica* Modulate Gut Microbiota Composition and Improve Short-Chain Fatty Acid Production

**DOI:** 10.3390/biology11101505

**Published:** 2022-10-14

**Authors:** Yun Zhang, Huan Chang, Shuai Shao, Lin Zhao, Ruiying Zhang, Shouwen Zhang

**Affiliations:** 1Postdoctoral Research Station, Heilongjiang Academy of Agricultural Sciences, Harbin 150086, China; 2College of Food Engineering, Heilongjiang East University, Harbin 150066, China

**Keywords:** *Opuntia ficus-indica*, anthocyanins, gut microbiota, high-throughput sequencing, SCFAs

## Abstract

**Simple Summary:**

*Opuntia ficus-indica* is rich in a variety of active substances, such as anthocyanins, flavonoids, and polysaccharides. Anthocyanins play an important role in regulating intestinal flora. To explore the relationship between anthocyanins in *Opuntia ficus-indica* and human intestinal flora, this study uses *Opuntia ficus-indica* as raw material to conduct animal experiments to study the effects of *Opuntia ficus-indica* anthocyanins on human gut microbes and short-chain fatty acid metabolites. The findings provide a theoretical basis for anthocyanins in *Opuntia ficus-indica* as dietary supplements to regulate human intestinal flora.

**Abstract:**

*Opuntia ficus-indica* is rich in a variety of active substances, such as anthocyanins, flavonoids, and polysaccharides. Some studies have shown that anthocyanins extracted from natural plants can regulate intestinal flora. The fruit was used as raw material, and anthocyanins were extracted from it. In vivo experiments were used to study the effect of *Opuntia ficus-indica* anthocyanins on the mouse intestine by 16S rRNA high-throughput sequencing (NovaSeq 6000 platform) and gas chromatography (hydrogen flame ionization detector (FID)) methods. Microbiota and effects of short-chain fatty acids (SCFAs). The results showed that after feeding anthocyanins, the diversity of intestinal microorganisms in mice was significantly increased (*p* < 0.05), the ratio of Firmicutes/Bacteroidetes (F/B value) was significantly decreased (*p* < 0.05), the relative abundances of beneficial bacteria Lactobacillus, Bifidobacterium, Prevotella, and Akkermansia in the intestinal tract of mice were significantly increased (*p* < 0.05), and the relative abundance of pathogenic bacteria Escherichia-Shigella and Desulfovibrio decreased significantly (*p* < 0.05). Furthermore, anthocyanins significantly increased the content of short-chain fatty acids in the cecum of mice, among which the content of acetic acid, propionic acid, and butyric acid increased the most. *Opuntia ficus-indica* anthocyanins can change the microbial diversity and flora composition of the mouse gut and promote the production of short-chain fatty acids. The findings provide a theoretical basis for the use of *Opuntia ficus-indica* anthocyanins as dietary supplements to regulate human intestinal flora.

## 1. Introduction

The gut microbiota comprise a group of bacteria that inhabit the human gut and exhibit interdependence with the human body. In the human stomach, there are a variety of microorganisms. From birth to aging, these intestinal microorganisms maintain a dynamic balance in the human body and restrict each other [1]. Because of the long-term synergy between the body and the intestinal flora, the gut microbiota forms an inseparable part of the human body and plays an immeasurably important role in maintaining physical health [2]. As a person grows, the intestinal flora also undergoes certain changes. Due to the influence of various acquired factors, such as living habits, psychological factors, mode of delivery, age, race, geographical environment, dietary habits (including consumption of probiotics), and antibiotics, and other factors that affect intake, unique microbial species are formed. Among these various factors, the main reason for the change in intestinal flora is diet, because it not only provides nutrients for the body, ut also provides energy for the flora, which changes the environment in which the flora lives. As a result, changing the diet structure may cause changes in the gut microbiota that allow the gut microbiota to return to normal, thereby improving health [3]. Diet is widely recognized as a major food-related factor affecting the composition and function of the human gut microbiota. For example, a diet rich in non-starch indigestible polysaccharides and dietary fiber affects the gut microbiota, primarily by reducing the Firmicutes/Bacteroidetes ratio [4].

The fruit produced by *Opuntia* plants is called *Opuntia ficus-indica*. *Opuntia* is a native plant found in the American continent. It is a multi-pulp plant, widely distributed in South Africa. It is also abundant in tropical areas of China, such as Hainan, Yunnan, and Guangxi, and has climate tolerance [5]. Its pulp contains a variety of bioactive substances, such as anthocyanins, polysaccharides, flavonoids, proteins, amino acids, and vitamins [6]. Anthocyanins are ingested by the human body and digested and absorbed by the gastrointestinal tract. The unabsorbed part of anthocyanins and their degradation products reach the area of the large intestine inhabited by microbes. The regulatory effect of anthocyanins on intestinal microorganisms is evident in two aspects. Anthocyanins interact with intestinal flora, and, through a series of metabolic activities, they have direct physiological activities (such as regulating diabetes and obesity). Further, anthocyanins and their degradation products change the number and composition of intestinal microorganisms by regulating the growth of specific bacteria in the microbial community and play a role in promoting human health [7]. Anthocyanins can also interact with endogenous and microbial enzymes, resulting in the production of a large number of circulating and excreted anthocyanin metabolites and catabolic products acting on the intestinal flora [8]. However, to the best of our knowledge, little is known about the functional interactions between *Opuntia ficus-indica* anthocyanins and the gut microbiota and their relevance to gut health. Therefore, the aim of this study was to investigate the effect of *Opuntia ficus-indica* anthocyanins on the gut microbiota and the production of short-chain fatty acids (SCFAs).

According to the previous in vitro fermentation study, we found that *Opuntia ficus-indica* anthocyanins are digested in an amount in the stomach, a large amount is digested in the small intestine, and most of them are left. They can change the diversity and composition of microbes in the intestinal tract and promote the production of short-chain fatty acids. In order to further confirm whether *Opuntia ficus-indica* anthocyanins can play a prebiotic role in the intestinal tract of normal mice, we took the animal experiment in vivo [9].

## 2. Materials and Methods

### 2.1. Materials and Reagents

*Opuntia ficus-indica* was obtained from Hainan Province, China, and *Opuntia ficus-indica* anthocyanins were purified from peeled *Opuntia ficus-indica*, and the purity is 14.16% [10]. HiPure StooL DNA Kits were acquired from Magen Biotechnology Co., Ltd. (Guangzhou, China). Agarose was purchased from Biowest. Goldview I was purchased from USA. PCR-related reagents were purchased from New England Biolabs (Ipswich, MA, USA). A mixed standard of 6 water-soluble fatty acids was purchased from Tanmo Quality Inspection Technology Co., Ltd. (Jiangsu, China). All other reagents provided by the laboratory were of analytical grade.

### 2.2. Animal Experiment

Forty male Kunming mice (body weight (BW) 20 ± 2 g) were purchased from Liaoning Changshen Biotechnology Co., Ltd. (SCXK(liao)2020-0001, Benxi, China). The animals were kept at relative humidity (60 ± 10%) and temperature (23 ± 2 °C) under controlled conditions, housed in a specific pathogen-free animal room (SPF grade) with a 12-h light–dark cycle (8 am to 8 pm), with ad libitum access to food and water. After 7 days of normal diet, the mice were randomly divided into four groups [11]: (1) K group (blank group), fed with sterile water every day; (2) L group (low dose group), fed with 50 mg/kg.BW anthocyanin solution every day; (3) M group (medium dose group), fed daily 100 mg/kg.BW anthocyanin solution; (4) H group (high dose group), fed 150 mg/kg.BW anthocyanin solution every day. Breeding procedures were strictly in accordance with national regulations.

### 2.3. Animal Behavioral Observation

The body weights of all mice were measured during the gavage period (the amount of each gavage was calculated). The daily mental state, coat color, gloss, food intake, water intake, and urine output of the mice were recorded. The fecal properties of the mice were recorded, and the samples were stored at −80 °C for further integrated analysis.

### 2.4. Calculation of the Coefficient of Each Tissue Organ

Sixteen hours after the last gavage, the mice were sacrificed by decapitation for aseptic sampling. The sampling sites were heart, liver, spleen, lung, kidney, thymus, and cecum contents (the contents of the cecum were stored in two parts). All tissue samples were weighed and frozen. After wrapping with tin foil, it was transferred to a sterile enzyme-free tube, pre-frozen in liquid nitrogen, and immediately transferred to a −80 °C refrigerator for storage.

The fat on the surface of the heart, thymus, spleen, liver, lung, and kidney was removed and washed with normal saline, the water on the surface was absorbed, and samples were weighed, organ weights were recorded, and the organ coefficient was calculated.

### 2.5. DNA Extraction

Genomic DNA was extracted using HiPure Stool DNA Kits according to the manufacturer’s instructions. By using a Nanodrop spectrophotometer (NC2000; Thermo Scientific, New York, USA), we compared the absorbance ratio A260/A280 to assess the purity of isolated DNA. DNA quality was confirmed by electrophoresis on a 0.8% agarose gel, and bands were visualized with a DYY-6C electrophoresis analyzer (Liuyi Instrument Factory, Beijing, China).

### 2.6. 16S rDNA Gene Amplification and Sequencing

Using forward primer 341F: CCTACGGGNGGCGWGCAG and reverse primer 806R: GGACTACHVGGGTATCTAAT, the target fragment 16S rDNA V3-V4 region was amplified by PCR. We amplified the target product with the barcode in the first primer as a specific primer. The first and second rounds of amplification were used to purify the PCR products using AMPure XP Beads (Beckman Coulter, Pasadena, CA, USA), and quantification was performed with Qubit3.0 (Thermo Fischer Scientific, Waltham, MA, USA) after purification. Using AMPure, the second-round amplification products were purified by XP Beads, quantified by ABI StepOnePlus Real-Time PCR System (Life Technologies, Grand Island, NY, USA), and sequenced on-board according to the PE250 mode pooling of NovaSeq 6000.

### 2.7. Bioinformatics Analysis of Gut Microbiota Profiles

The UCLUST function in QIIME was used to select high-quality sequence data and cluster them into operational taxonomic units (OTUs) with 97% similarity. Bacteria were then determined using the number of OTUs observed α Diversity. In QIIME and R software, unweighted UniFrac principal coordinate analysis (PCoA), unweighted UniFrac non-metric multidimensional scale (NMDS), and unweighted pairwise group method (UPGMA) with arithmetic mean were used for β diversity analysis. The potential microbial communities of different regions and species were further compared between the groups using transfer data.

### 2.8. Detection of Short-Chain Fatty Acids in the Cecum by Gas Chromatography

#### 2.8.1. Treatment of Cecal Contents

We took 100 mg of cecal contents into a test tube, added 5 mL of ultrapure water to an ice-water bath, vortexed to mix the samples, then sonicated for 10 min, and immediately placed them in an ice-water bath for 20 min. Sampled were then centrifuged at 4800 r/min for 20 min. The supernatant was passed through a 0.22-μm filter and transferred to a gas chromatography injection bottle for testing [12].

#### 2.8.2. Gas Chromatography Conditions

The initial oven temperature was 70 °C for 1 min, then increased to 160 °C at 15 °C/min and to 210 °C at 30 °C/min, and finally held at 210 °C for 5 min. The sample loading volume was 1.5 μL, the chromatographic column was a 10 m DB-WAX capillary column, the injection port temperature was 220 °C, the split ratio was 10:1, the column flow rate was 1.5 mL/min, and the nitrogen cross-flow was performed using a hydrogen flame detector (FID) at a detector temperature of 250 °C [13,14].

### 2.9. Statistical Analysis

The paired sample test was used for statistical evaluation, and SPSS 17.0 analysis software was used. A *p* value of <0.05 was considered to indicate statistically significant results. All bar graphs in this study were exported using Origin 2021. The data were expressed as mean ± SD. NovaSeq 6000 was used for gut microbiota diversity analysis.

## 3. Results

### 3.1. Effect of Opuntia ficus-indica Anthocyanins on the Body Weight of Mice

The body weight changes of all Kunming mice in the experiment reflected their physical condition, as shown in Figure 1. After one week of acclimatization, all mice had different body weights due to individual differences between mice. The weights of group K were in the range of 38.7–42.3 g, that of group L was 35.9–39.4 g, that of group M was 34.1–38.6 g, and that of group H was 40.6–43.6 g. There was no significant difference in the body weights of mice between groups (*p* < 0.05). The weights of mice in each group showed an upward trend during the feeding period. The increase was relatively slow after 15 days, and there was no significant difference compared with the K group (*p* < 0.05). The results showed that *Opuntia ficus-indica* anthocyanins did not affect the body weights of normal mice. During the feeding period, the mental state, coat color, gloss, and feces of the mice were compared with those of the normal mice. The mice urinated and defecated normally, indicating that the anthocyanins from the *Opuntia ficus-indica* did not affect the physiological state of the normal mice.

### 3.2. Effects of Opuntia ficus-indica Anthocyanins on the Organ Coefficient of Mice

Organ coefficient is the ratio of organ weight to body weight of experimental animals. Excluding the difference in the intragastric dose, under all the same conditions such as feeding and ignoring the individual differences in the mice, the coefficient of each organ can well reflect the physical condition of the mice. It can be seen from Table 1 that after the three-week gavage, there were no significant differences in the heart coefficient, liver coefficient, spleen coefficient, lung coefficient, and kidney coefficient of the mice in the L, M, and H groups compared with that in the K group (*p* > 0.05); thus indicating that the administration of anthocyanins will not cause adverse reactions to various organs of normal mice, such as organ enlargement or shrinkage.

### 3.3. PCR Amplification Gel Electrophoresis Results

PCR amplification was performed on the 16SrDNAV3-V4 region of the target fragment. As can be seen from Figure 2, the bands were clear and the size was correct, all of which were a single band, and the next test could be carried out.

### 3.4. α-Diversity

Compared to the K group, the values of Chao1 index and ACE index decreased significantly (*p* < 0.01) after three weeks of *Opuntia ficus-indica* anthocyanin feeding, in a dose-dependent manner, indicating a change in the uniformity of gut microbes (Table 2). Compared with the K group, the Shannon index and Simpson index increased significantly (*p* < 0.05), indicating that the diversity of the flora had been altered (Table 2). The results show that the intestinal bacteria of mice fed *Opuntia ficus-indica* anthocyanins had greater variation, which was the same effect as observed following the previous in vitro fermentation of *Opuntia ficus-indica* anthocyanins. The curve of sequencing depth at 20,000 was flattened out (Figure 3), indicating that the sequencing was reliable. It can be seen from Figure 4 that the blank group has the longest curve and a uniform decline, indicating that the abundance of the sample composition is relatively uniform and the species composition diversity is high. Compared with the blank group, the curves for the low, medium, and high dose groups decreased rapidly and steeply, indicating that the distribution of species abundance in the samples was low, the proportion of dominant bacteria was high, and the diversity was reduced. The curves of the middle and high dose groups were similar, indicating that the species similarity was high, and it was further concluded that the low dose group contained the highest dominant flora. The results indicate that feeding anthocyanins to mice could change the species composition richness and increase the growth of dominant flora.

### 3.5. β-Diversity

The results from the analysis of differences between the PCoA samples are shown in Figure 5a. The points of color do not overlap or intersect with each other, indicating that the samples are completely separated, which can better describe the characteristics of the mouse intestine. The results indicate that the microbial colonies in the guts of the mice were altered and varied. Among them, the distance between group H and group K is the largest, indicating that the similarity between group H and K is the lowest, i.e., the similarity of the microbial community is the lowest and the difference is large. Furthermore, the distance between group L and group K is the smallest, indicating that group L and K have the greatest similarity. The difference is small. From Figure 5b, NMDS can obtain the stress function value streets = 0.014 ≤ 0.01, which indicates that it has excellent representativeness. From Figure 5c, it can be concluded that R = 0.633 indicates a significant difference between groups, and *p* = 0.001 < 0.05 is significant. According to the results of the ANOSIM analysis, R = 0.985 indicates a significant difference between groups, and *p* = 0.001 < 0.01 is extremely significant.

### 3.6. OTU Numbers

There are four groups, namely group K, group L, group M, and group H. There are 12 samples in total, with three samples in group. The average sequencing depth of these 12 samples is 63752 ± 2235. According to the similarity of 97%, each sample has 905 ± 55 OTUs, indicating that the measurement results can accurately represent the real situation of the intestinal flora in mice. Groups L, M, and H shared 35 OTUs, accounting for 1.4% of the total sequence. There were 411 OTUs in group L, group M, group H, and group K, accounting for 16.2% of the total sequence. The unique OTUs of groups K, L, M, and H were 455, 358, 411, and 510, respectively, accounting for 17.9%, 14.1%, 16.2%, and 20.1% of the total sequences, respectively (Figure 6). The results showed that compared with that in group K, *Opuntia ficus-indica* anthocyanins altered the composition of the cecal microbiota in mice.

### 3.7. Effect of Opuntia ficus-indica Anthocyanins on Microbiota Taxonomic Composition

#### 3.7.1. Species Composition Analysis at the Phylum Level

As shown in Figure 7, the intestinal microbes of mice mainly include Firmicutes, Bacteroidetes, Proteobacteria, and Patescibacteria, in addition to a small number of warts. Compared with the K group, Proteobacteria decreased significantly (*p* < 0.05), Bacteroidetes increased significantly (*p* < 0.05), and Firmicutes also showed an increasing trend. Compared with the K group, the ratios of Firmicutes/Bacteroidetes decreased to 40.2%, 21.4%, and 14.1%, respectively (Figure 8). These results suggest that *Opuntia ficus-indica* anthocyanins alter the composition of the gut microbiota in mice.

#### 3.7.2. Species Composition Analysis at the Genus Level

*Desulfovibrio*, *Ruminococcaceae_UCG-014*, and *Candidatus_Saccharimonas* were in group K. After the oral administration of anthocyanins, *Prevotella*, *Akkermansia*, *Lactobacillus*, and *Bifidobacterium* were the main species (Figure 9). After the oral administration of anthocyanins, compared with the K group, the proportion of *Escherichia coli* was significantly decreased (*p* < 0.01), the proportion of *Desulfovibrio* was significantly decreased (*p* < 0.05), and the proportion of *Prevotella* and *Lactobacillus* significantly increased. (*p* < 0.05). *Bifidobacteria* also showed an upward trend (Figure 10). In addition, compared with the K group, *Akkermansia* was unique in the L, M, and H groups. These results suggest that gavage with *Opuntia ficus-indica* anthocyanins is beneficial for the growth of intestinal probiotic-producing bacteria in mice.

### 3.8. Effect of Opuntia ficus-indica Anthocyanins: SCFA production

As shown in Figure 11, the concentrations of acetic acid, propionic acid, and butyric acid in *Opuntia ficus-indica*-anthocyanins-treated groups increased relative to the control (*p* < 0.05). Compared with the K group, the contents of i-butyric acid, i-valeric acid, and valeric acid were significantly different in the M and H groups (*p* < 0.05). It was also found that the *Opuntia ficus-indica*-anthocyanins-treated groups exhibited significantly higher production of total SCFAs relative to the K group (*p* < 0.05). The results revealed that *Opuntia ficus-indica*-anthocyanins can be fermented to SCFA-producing bacteria.

## 4. Discussion

Among the many effects of anthocyanins, the most popular among scholars is the antioxidant effect [15]. However, in recent years, studies have found that anthocyanins can play a role in promoting human health by regulating intestinal flora. For example, in some immune diseases such as allergies, cardiovascular and cerebrovascular diseases, such as high blood pressure, high cholesterol, high blood sugars, and thrombosis, as well as some chronic diseases and metabolic diseases, such as diabetes; it also has a protective effect on the nervous system and the prevention of obesity [16,17,18,19,20,21]. However, according to previous in vitro simulated digestion results from our research group, the digestibility of *Opuntia ficus-indica* anthocyanins through the stomach intestine was 34.9%, with 65.1% finally reaching the large intestine. This shows that most of the anthocyanins of *Opuntia ficus-indica* will not be digested by the human body and may reach the large intestine and be utilized by the intestinal flora [22,23,24]. Therefore, starting from the regulation of intestinal flora, preventing the occurrence of certain diseases has become a research hotspot [25]. The gut microbiota is considered to be an essential part of maintaining human health, with protective effects against the pathogenesis of various diseases [26]. Wang W et al. [27] used *Lactobacillus plantarum* 69-2 and GOS supplements in aging model mice, and the results showed that *L. plantarum* 69-2 and GOS could activate the hepatic AMPK/SIRT1 signaling pathway by regulating the gut microbiota and metabolites through the liver-gut axis to restore hepatic antioxidant activity to alleviate aging. Due to the digestion of certain compounds, such as polyphenols and polysaccharides, in the gastrointestinal tract, these compounds can interact with the gut microbiota to alter gut bacterial diversity [28]. Gowd et al. [29] studied the degree of digestion and absorption of anthocyanins by establishing an in vitro digestion model to simulate oral and gastrointestinal digestion. The results showed that anthocyanins interacted with intestinal bacteria to present a dose gradient. Accordingly, it was inferred that the specific source of anthocyanins has an influence on their digestion and absorption. Our study also showed that *Opuntia ficus-indica* anthocyanins can modulate the composition of gut microbiota.

The abundance of bacterial species among the gut microbiota was determined based on OTUs. Some scholars found that the polysaccharides in purple potato increased the number of OTUs and α-diversity index in normal mice [30]. Consistent with these findings, *Opuntia ficus-indica* anthocyanins increased the number of OTUs in fecal bacteria in mice. The intestinal microbes of mice mainly include Firmicutes, Bacteroidetes, Proteobacteria, and Patescibacteria, in addition to a small amount of Verrucomicrobia, Actinobacteria, and Epsilonbacteraeota. Among them, Bacteroidetes and Firmicutes are the two most important phyla in the gut, accounting for nearly 90% of the total number of bacteria [31]. Bacteroidetes are gram-negative bacterial phyla that play an important role in carbohydrate metabolism. Mammalian gut Bacteroides have SUS-like systems that target a variety of different glycans [32]. Firmicutes, a diverse group of Gram-positive bacteria, are considered to be the major butyrate producers and major degraders of indigestible polysaccharides in the gut [33]. The present study found that Bacteroidetes and Firmicutes in the intestinal tract of mice showed an upward trend after gavage with *Opuntia ficus-indica* anthocyanins. In contrast, Proteobacteria, which represent the vast majority of pathogenic bacteria, showed a downward trend [34,35,36].

Wang W et al. [37] found that when healthy mice were fed *Isaria cicadae Miquel* (ICM) fruiting body polysaccharides, the relative abundances of *Lactobacillus*, *Akkermansia*, and *Bacteroides* were found to increase significantly, while that of *Clostridium* decreased significantly. Some researchers have used black raspberry anthocyanins as a dietary supplement to enhance the growth of *Eubacterium rectum*, *Faecalibacterium prausnitzii*, and *Lactobacillus* and inhibit the growth of *Desulfovibrio* and *Enterococcus* [38]. Wang et al. [39] showed that cyanidin-3-O-glucoside and black rice anthocyanins significantly induced a significant increase in the number of *Bifidobacteria* and *Lactobacilli* (*p* < 0.05). It has the effect of regulating intestinal microbes to stimulate the growth of beneficial bacteria. Our results also showed that the number of *Lactobacillus*, *Bifidobacterium*, and *Prevotella* in the intestinal tract of mice increased after gavage with anthocyanins from *Opuntia ficus-indica*. In addition, the unique *Akkermansia* bacteria were identified. As a new type of probiotic, *Akkermansia* can maintain a stable state of intestinal microbes and reduce the occurrence of obesity, diabetes, intestinal inflammation, and liver disease. Muriel et al. [40] found that in patients with gastrointestinal inflammation and metabolic disorders, *Akkermansia* was generally reduced in patients. From this, it was inferred that *Akkermansia* had some anti-inflammatory effect. Plovier [41] found that *Akkermansia* was reduced in diabetic obese mice fed high-fat diet, fat mass development, insulin resistance, and dyslipidemia in mice. Moreover, the content of *Desulfovibrio* and *Escherichia coli* decreased after gavage with *Opuntia ficus-indica* anthocyanins. *Desulfovibrio* can release a large amount of hydrogen sulfide (H_2_S) in the process of metabolism. Therefore, in recent years, it has been proposed that *Desulfovibrio* can damage the intestinal epithelium, thereby causing lesions of the digestive system [42,43,44]. Therefore, the current results show that anthocyanins from *Opuntia ficus-indica* promoted the growth of beneficial bacteria and inhibited the growth of harmful bacteria.

Studies have shown that during the gastrointestinal digestion of anthocyanins in purple cabbage, bioactive compounds can be metabolized by human colonic flora to produce metabolites with higher biological activity and more beneficial effects; these metabolites are known as short-chain fatty acids (SCFAs). Our results also showed that *Opuntia ficus-indica* anthocyanins can up-regulate the levels of acetate, propionate, and butyrate in mice. In addition to providing energy for intestinal epithelial cells, SCFAs play an important role in maintaining water and electrolyte balance, regulating intestinal pore balance, improving intestinal function and resistance to microorganisms, have anti-inflammatory properties, and preventive effects against obesity and type 2 diabetes mellitus (T2DM) [45]. In addition to being an important energy source for intestinal cells, acetic acid also activates G protein-coupled receptors, which activate fat-insulin signaling [46]. Acetic acid is one of the major metabolites of the gut, which not only reduces appetite by directly stimulating the nervous system, but also prevents obesity-related hyperinsulinemia and hypertriglyceridemia [47]. Propionate is involved in immune regulation and reduces high fatty acid levels in the liver and plasma [48]. Propionate also increases the number of gut-derived regulatory T cells and positively affects the central nervous system by increasing myelin regeneration [49]. Short-term rectal administration of propionate has been shown to improve depressive symptoms in rats in a chronic unpredictable mild stress (CUMS) model [50]. Butyrate stimulates the expression of fatty acid oxidation genes, thereby reducing total cholesterol in the liver [51]. Butyrate also increases the concentration of the central neurotransmitter 5-HT, promotes the expression of brain-derived neurotrophic factor (BDNF), and significantly improves depression-like behavior in CUMS model mice [52]. Our results show that *Opuntia ficus-indica* anthocyanins are fermented to produce various SCFAs, suggesting that the intake of *Opuntia ficus-indica* anthocyanins is beneficial to health.

## 5. Conclusions

Anthocyanins from *Opuntia ficus-indica* can play a role in regulating the intestinal flora of mice, reducing pathogenic bacteria and increasing beneficial bacteria. In addition, the intake of anthocyanins from *Opuntia ficus-indica* can modulate SCFA-producing bacteria, thereby increasing the content of total SCFAs. Our findings provide new ideas for the potential dietary application of anthocyanins from *Opuntia ficus-indica*.

## Figures and Tables

**Figure 1 biology-11-01505-f001:**
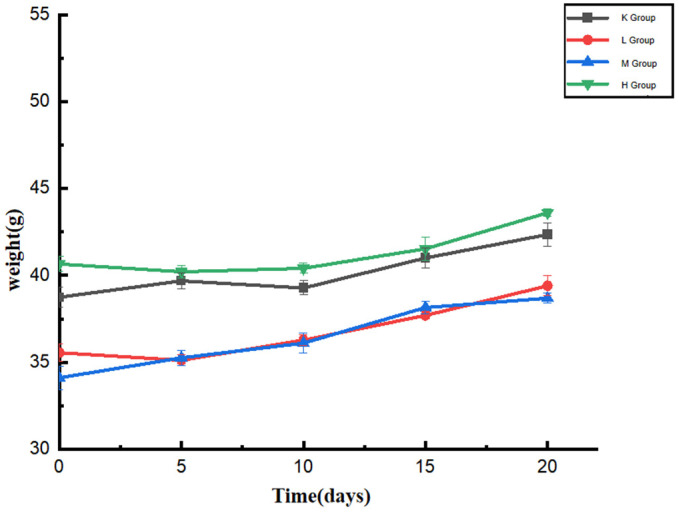
Changes in the body weights of mice during oral administration of anthocyanins from *Opuntia ficus-indica*.

**Figure 2 biology-11-01505-f002:**
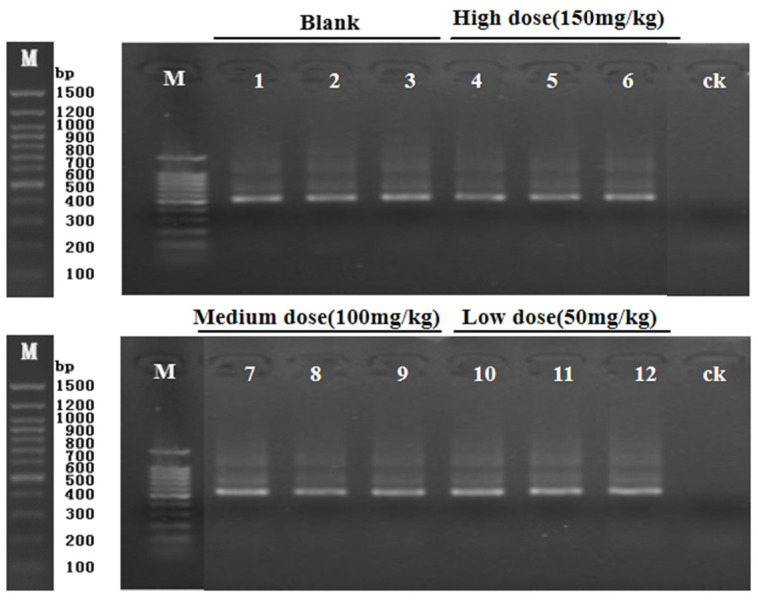
Electrophoresis of cecal contents of mice. PCR products 1, 2, and 3 represent K1, K2, and K3 (blank); 4, 5, 6 represent H1, H2, and H3 (high dose); 7, 8, and 9 represent M1, M2, and M3 (medium dose); 10, 11, and 12 represent L1, L2, and L3 (low dose).

**Figure 3 biology-11-01505-f003:**
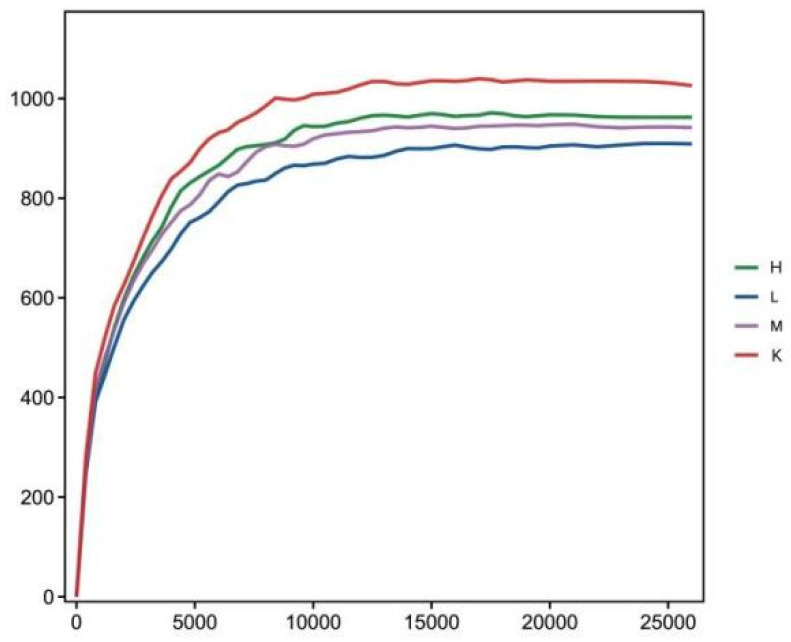
Sample rarefaction curve.

**Figure 4 biology-11-01505-f004:**
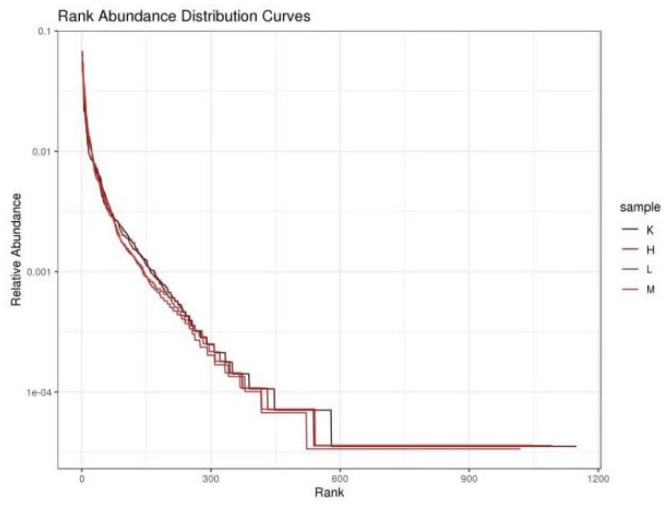
Species Abundance Rank Curve.

**Figure 5 biology-11-01505-f005:**
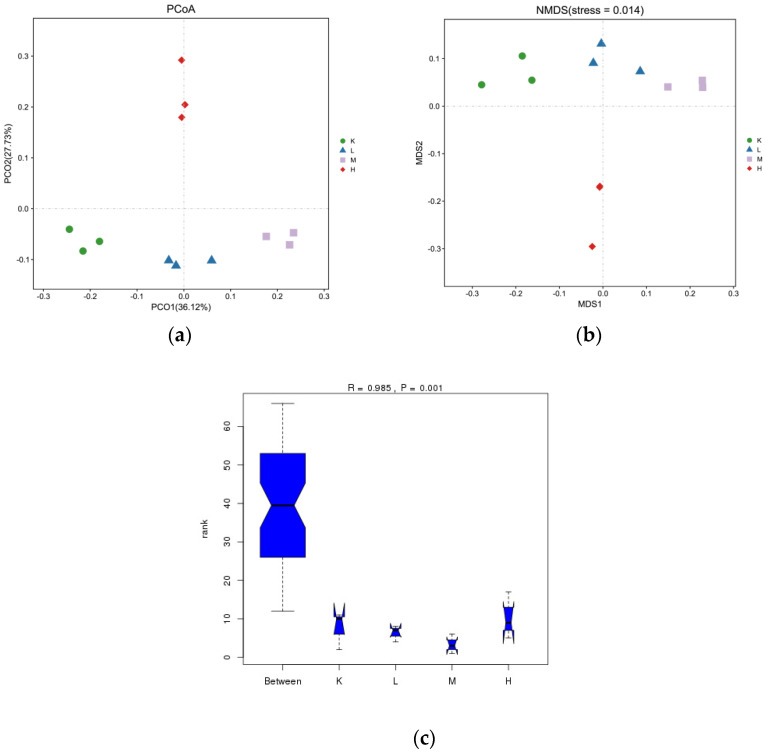
Anthocyanin microbial flora in *Opuntia ficus-indica*.

**Figure 6 biology-11-01505-f006:**
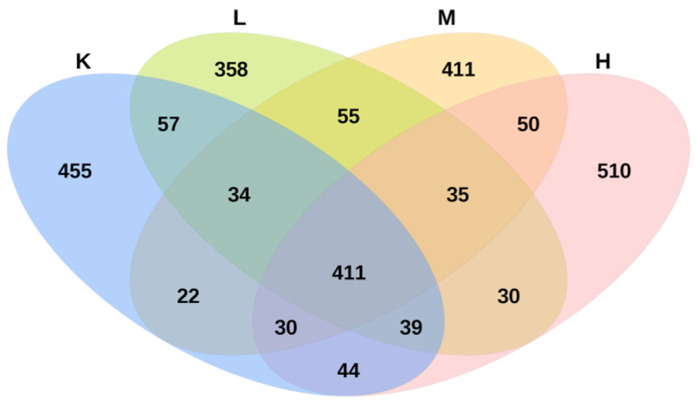
Species enrichment Venn diagram.

**Figure 7 biology-11-01505-f007:**
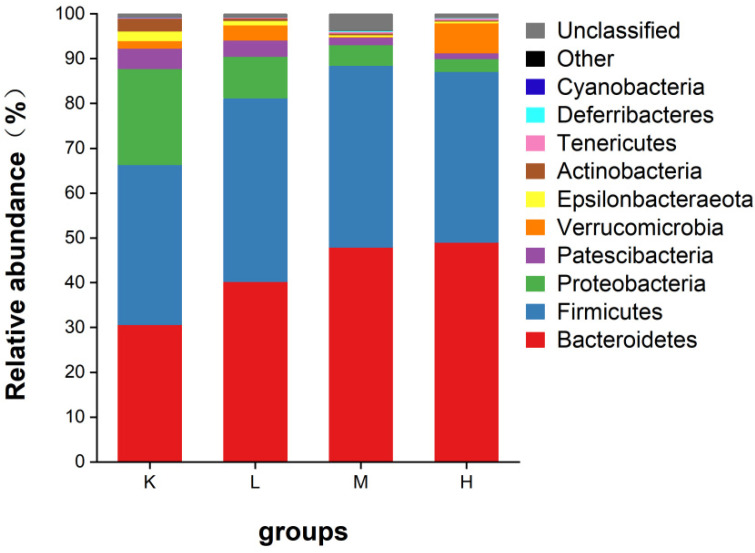
Relative abundance of gut microflora phylum levels in mice.

**Figure 8 biology-11-01505-f008:**
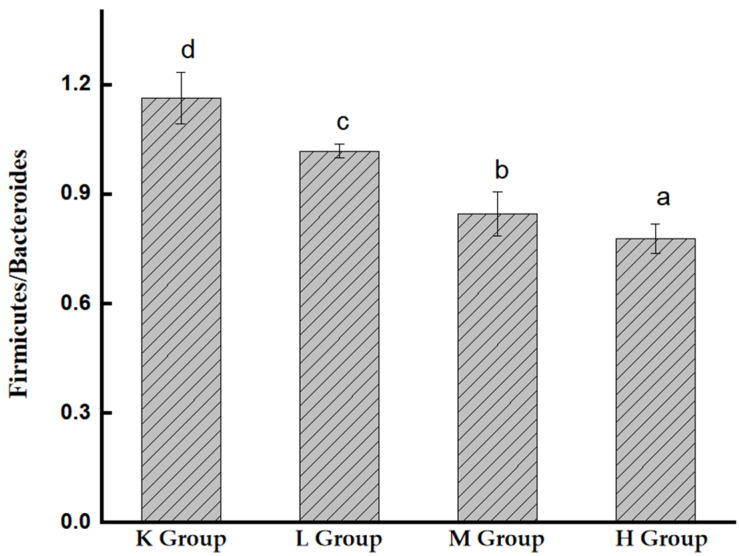
The ratio of Bacteroidetes/Firmicutes in the intestinal flora of mice. Different lowercase letters indicate significant differences (*p* < 0.05).

**Figure 9 biology-11-01505-f009:**
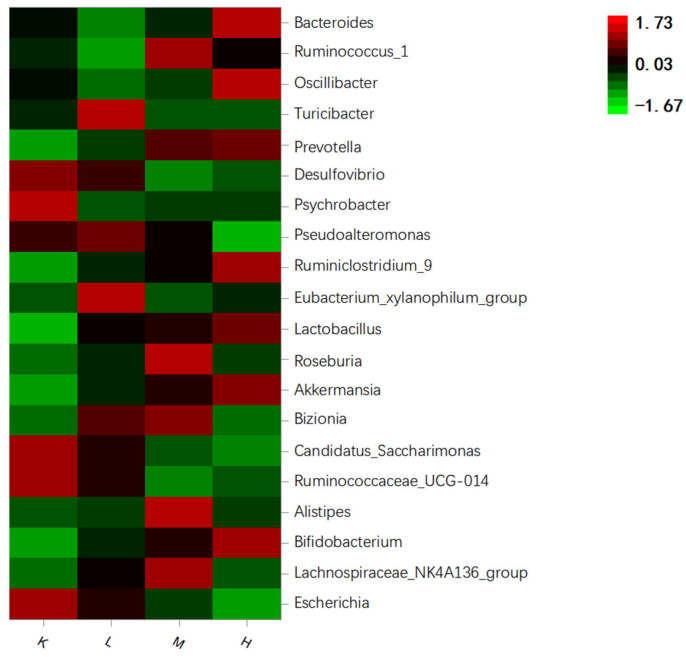
Heat map of species enrichment of gut microbiota in mice.

**Figure 10 biology-11-01505-f010:**
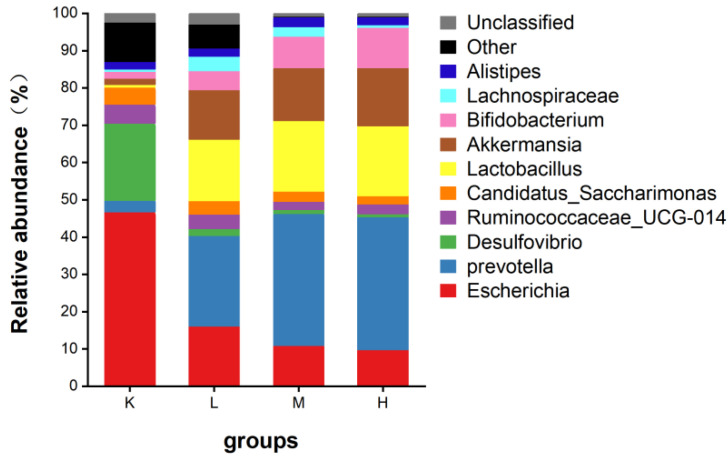
Relative abundance of gut microflora genus levels in mice.

**Figure 11 biology-11-01505-f011:**
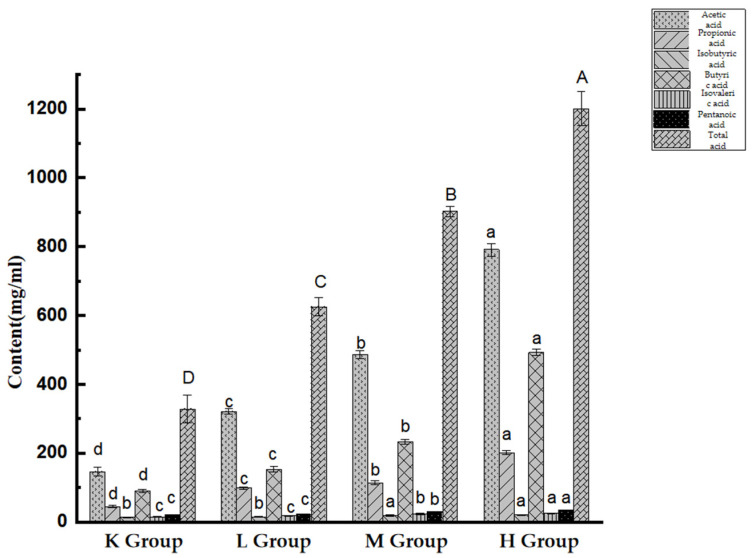
Content of short-chain fatty acids within mouse cecal contents. Different lowercase letters indicate significant differences between groups (*p* < 0.05), and different capital letters indicate extremely significant differences between groups (*p* < 0.01).

**Table 1 biology-11-01505-t001:** Mouse Organ Ratio Index.

Group	Cardiac Coefficient%	Liver Coefficient%	Spleen Coefficient%	Lung Coefficient%	Kidney Coefficient%
K group	0.487 ± 0.04	4.663 ± 0.38	0.303 ± 0.08	0.586 ± 0.08	1.447 ± 0.19
L group	0.584 ± 0.07	4.661 ± 0.58	0.337 ± 0.11	0.698 ± 0.12	1.599 ± 0.16
M group	0.529 ± 0.04	4.742 ± 0.58	0.344 ± 0.06	0.744 ± 0.19	1.456 ± 0.11
H group	0.543 ± 0.06	4.844 ± 0.46	0.387 ± 0.21	0.638 ± 0.11	1.439 ± 0.18

**Table 2 biology-11-01505-t002:** Effects of anthocyanins on the alpha diversity index of gut microbiota.

Group	Chao1	ACE	Shannon	Simpson
K group	1023.41 ± 43.58	2052.29 ± 48.65	7.08 ± 0.07	0.98 ± 0.004
L group	905.81 ± 9.51 **	929.96 ± 13.27 **	7.53 ± 0.04 *	1.53 ± 0.001 *
M group	938.84 ± 37.39 **	967.38 ± 40.00 **	7.82 ± 0.03 *	1.95 ± 0.001 *
H group	958.26 ± 32.89 **	987.78 ± 32.94 **	8.31 ± 0.18 *	2.42 ± 0.003 *

Note: * and ** indicates a statistically significant difference compared with the blank group, * *p* < 0.05, ** *p* < 0.01.

## Data Availability

All data dealing with this study are reported in the paper.

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
