# Peer review of "Anthocyanins from Opuntia ficus-indica Modulate Gut Microbiota Composition and Improve Short-Chain Fatty Acid Production"

_biology, 2022, doi:10.3390/biology11101505_

Round 1

Reviewer 1 Report

The overall manuscript is well done, although the authors show that anthocyanins promote changes in the microbiota composition the findings are somehow speculative on the potential health benefits.

The manuscript could be improved significantly if the authors show a real outcome in high-fat feed mice. Furthermore, the mouse model used " Kunming mice" is not known to be responsive to any metabolic challenge. 

Reviewer 2 Report

This article "Anthocyanins from Opuntia ficus-indica modulate gut microbiota composition and improve short-chain fatty acid production" was revised and has a novelty and I recommend it for publication after consideration of the following comments.

1. The purity of the extracted anthocyanins was not mentioned in the study, and it is recommended that the purity be supplemented.

2. Should the title be “2.6. 16S rDNA” on line 127?

3. In section 2.6, the sequences of primers should be clearly explained.

4. The footnote to Table 1 shows: "* indicates a statistically significant difference compared with the blank group", but does not show "*" in the table. Footnotes can be deleted if the difference is not significant. Also, no significant difference should be "p > 0.05" on the line 189.

5. Why there is no footnote in Table 2, the explanation about "*". Also, "p<0.05" and "p<0.01" should be represented by different symbols

6. Both "Fig." and "Figure" appear in the manuscript, and it is recommended to unify the whole text.

7. Line 250, an "each" should be removed.

8. There is a description of "(p < 0.05)" in Section 3.7.1, but no significant symbols are marked in Figure 7.

9. The Latin name of the gut microbiota should be italicized.

10. Why is Figure 8 absent in the manuscript?

11. Some recent references are suggested to be added to the discussion, as follow:

Wang W, Xu C, Liu Z, et al. Physicochemical properties and bioactivity of polysaccharides from Isaria cicadae Miquel with different extraction processes: effects on gut microbiota and immune response in mice[J]. Food & Function, 2022. https://doi.org/10.1039/D2FO01646J

Wang W, Liu F, Xu C, et al. Lactobacillus plantarum 69-2 combined with galacto-oligosaccharides alleviates d-galactose-induced aging by regulating the AMPK/SIRT1 signaling pathway and gut microbiota in mice[J]. Journal of Agricultural and Food Chemistry, 2021, 69(9): 2745-2757.

12. English language advice should be more professional.

Reviewer 3 Report

In this paper, the Kunming mouse model was used to observe the regulatory effect of anthocyanins from Opuntia ficus-indica on gut microbiota and short-chain fatty acids. It providing a theoretical basis for Opuntia ficus-indica anthocyanins as dietary supplements to regulate the health of human gut microbiota. The specific comments are as follows:

1. The research with reasonable design, correct statistical methods and credible data. It has theoretical significance and practical value. But the purity of anthocyanins is not shown in this paper.

2. The identification of the “control group” and the “blank group” in the article is not clear enough, it is not marked in the previous article.

3. There are also some formatting issues. The number sizes in “Figure 2” are not uniform, and the size of figures is obviously different.

4. The paper refers to a wealth of literature, but the format of references is irregular. Need to be revised in accordance with submission requirements

5. The language description of some parts of the article is not fluent enough. There are also some syntax errors in the article.

Round 2

Reviewer 1 Report

Thanks for the answers. 

Looking forward to reading the next study on high-fat diet feed mice.